# Carbon Monoxide (CO) as a Retinal Regulator of Heme Oxygenases -1, and -2 (HO’s) Expression

**DOI:** 10.3390/biomedicines10020358

**Published:** 2022-02-01

**Authors:** Sławomir Nowak, Przemysław Gilun, Katarzyna Kozioł, Maria Romerowicz-Misielak, Magdalena Koziorowska-Gilun, Barbara Wąsowska

**Affiliations:** 1Department of Biotechnology, Institute of Biology and Biotechnology, College of Natural Sciences, University of Rzeszow, Pigonia 1, 35-310 Rzeszów, Poland; katko@ur.edu.pl (K.K.); mromerowicz@ur.edu.pl (M.R.-M.); 2Department of Local Physiological Regulations, Institute of Animal Reproduction and Food Research of the Polish Academy of Sciences in Olsztyn, Tuwima 10 Str., 10-748 Olsztyn, Poland; b.wasowska@pan.olsztyn.pl; 3Department of Animal Biochemistry and Biotechnology, Faculty of Animal Bioengineering, University of Warmia and Mazury, Oczapowskiego 5, 10-719 Olsztyn, Poland; magda.koziorowska@uwm.edu.pl

**Keywords:** heme oxygenase -1 and -2, carbon monoxide, retina, light signal

## Abstract

Carbon monoxide (CO) has been proposed as a chemical light signal and neural system modulator via heme oxygenases -1 and -2 (HO-1 and HO-2). Many papers have proven the CO-HO circuit to be important for such physiological pathways as the molecular biological clock and the GnRH axis, but also in such pathological occurrences as ischemic injuries, or inflammation as a regenerative and neuroprotective factor. In this in vivo experiment, we used three groups of pigs: control—housed in natural conditions without any procedures; without CO—adapted and kept in constant darkness, infused with blank plasma; and with CO—adapted and kept in constant darkness infused with CO-enriched plasma. After the experiments, each animal was slaughtered and its eyes were collected for further analysis. Quantitative PCR and Western blot analysis were performed to show statistical differences in the expressions between the experimental groups. Our data revealed that exogenous CO is regulator of mRNA transcription for *HO-1* and *HO-2* and *PCNA*. Moreover, the mRNA abundance of analyzed factors in the experimental group after CO elevation revealed a restored gene-expression level similar to the control group, which we had observed in the group’s restored protein level after CO elevation. In conclusion, exogenous CO regulates *HO’s* and *PCNA* gene expression on transcriptional and translational levels in a similar way as a light cue.

## 1. Introduction

The retina tissue of the eye consists of rods and cone nerve endings which are responsible for sending information concerning light intensity and colour after processing by downstream retinal neurons (bipolar, horizontal, amacrine and ganglion cells). Light signals are transmitted to the brain via the axons of ganglion cells for further analysis. This part of the retina is mainly responsible for the transmission of the image-forming signal c. In the past few years, scientist have identified small subsets of retinal ganglion cells which are intrinsically photosensitive. This part is responsible for the transmission non-image-forming light signal through the use of photo pigment melanopsin [1]. These structures are the most sensitive to factors of a changing environment and contribute to increased or decreased HO enzyme and carbon monoxide (CO) production. It is important to determine whether other factors, not related to the light intensity and length of exposition, can affect the changes in the synthesis of the HO enzyme in the eye The heme oxygenase (HO) is the main enzyme which catalyses heme degradation to Fe^2+^, biliverdin and carbon monoxide (CO). HO recycles free iron from senescent red blood cells in animals. Free iron activates the expression of both iron-binding proteins—ferritin and Fe-ATPase (an iron transporter)—which decrease Fe^2+^ concentrations in cells. Biliverdin is the second product of HO catalysis, which is further converted to bilirubin. Carbon monoxide (CO) is the last product of heme degradation and is removed through the lungs during respiration [2]. The main CO transporter, carboxyhaemoglobin, persists in the body at levels of 1–3%, but such concentrations do not have a toxic effect [3]. CO at low concentrations, physiological or slightly raised, is an anti-inflammatory and anti-apoptotic factor in body tissues such as the lungs [4] and blood vessels [5], while HO-1 deficiency disturbs iron metabolism and redistribution, which causes microcytic anaemia [6].

Of the three HO enzyme isoforms, HO-1 and HO-2 are different but actively catalytic, while little is known about the elusive HO-3 [7]. The HO-1 enzyme, with a molecular weight of 32 kDa and inducible activity, belongs to the heat shock protein family (group HSP32) and is highly expressed in the spleen and liver tissues of mammals. *HO-1* gene expression is dependent on reactive oxygen species (ROS), reactive nitrogen species (RNS), heavy metals, UV radiation, hypoxia and various chemicals. Induction of the HO-1 enzyme with curcumin, caffeic acid phenethyl ester, cobalt protoporphyrin IX and dimethyl fumarate leads to the removal of ROS from the body [8,9]. HO-1 deficiency affects iron metabolism and redistribution which lead to microcytic anaemia [6]. The HO-2 isoform has a molecular mass of 36 kDa and is mainly constitutively expressed. Its high expression has been observed in the tissue of the kidneys, vasculature, brain and testes [10]. The HO-3 isoform, very similar to the HO-2 isoform but with a lower enzymatic activity, has a molecular mass of 33-kDa [11].

Carbon monoxide (CO) is a colourless, odourless, non-irritant gas that accounts for numerous cases of poisoning. After entering through the lungs, CO binds with hemoglobin to form carboxyhemoglobin (COHb), with an affinity 200 times greater than oxygen, which results in oxygen displacement. The first symptoms of CO poisoning are observed when COHb reaches a level of 15–30%; an increase in the level to 70% leads to unconsciousness and death [12]. The CO is also considered to be a neurotransmitter, whose presence and function are related to seasonal changes. Longer light exposure increases endogenous CO concentration, while a lack of light decreases it [13]. The retina is one of the structures involved in CO production and its release into the blood, and is also responsible for receiving light stimuli. The HO enzyme, under the influence of external factors, participates in the production of CO in the retina. The CO molecules penetrate the eye retina tissue to the ophthalmic venous blood, and then are transferred via the humoral pathway to the brain [14].

It was established that changes in CO concentration significantly impact the stimulation of such biological regulators as *GnRH* gene and *GnRH* receptor expression [15], as well as combating both inflammatory disorders [16] and endogenous anti-inflammatory, anti-apoptotic, and anti-proliferative effects [17]. The non-physiological changes in CO concentration in the blood have also been demonstrated to interfere with melatonin synthesis in the pineal gland tissue [18]. The published results show that the CO also modulates intraocular pressure and can be used to treat eye diseases [19]. Furthermore, exogenous CO has been shown to act as a promoter of neurogenesis and a suppressor of neuroinflammation and blood–brain barrier disruption [20].

The aim of this study was investigate whether the chemical particle CO as an artificial light signal triggers the natural synthesis of HO in the retina. The research includes the influence of CO on the proliferative capacity of cells in the structure of the retina. Proliferation capacity and HO will be tested by gene expression at mRNA and protein levels in the tissue.

## 2. Materials and Methods

### 2.1. Animals

The eyes from twenty-two domestic pigs (*Sus scrofa f. domestica*) were used for the experiments at the Department of Local Physiological Regulations, Institute of Animal Reproduction and Food Research, Polish Academy of Sciences in Olsztyn, Poland. The Local Ethics Committee in Olsztyn issued approval no. 13/2017 for animal testing on 28 February 2017. The obtained material was analyzed at the Department of Biotechnology, Institute of Biology and Biotechnology, University of Rzeszow, Poland. Twenty two 8–10-month-old domestic pigs were studied from March to April, when the length of the night in natural conditions was close to the length of the day. Before the start of the experiments, the animals in the experimental groups were adapted to complete darkness for forty-eight hours.

Control group (*n* = 6): The animals were killed in the middle of subjective 12.00 (12pm) (*n* = 6). Animals in this group were not subjected to any experiments and were kept under natural light conditions.

Without CO group (*n* = 8): The animals were kept in darkness for forty-eight hours, and then a cannula was inserted into the angular vein of the eye. For a further seventy-two hours they were in total darkness, and plasma infusion without the addition of CO was administered. After the infusion was completed, the slaughter was conducted as above (control group).

With CO group (*n* = 8): Animals were kept in darkness for forty-eight hours, and then a cannula was inserted into the angular vein of the eye. For a further seventy-two hours, they were in total darkness, and plasma was administered continuously with the addition of CO. After the infusion was complete, the slaughter was carried out as above (control group). Detailed description of the animal model is in Gilun [21].

Blood plasma was prepared prior to infusion. We elevated CO in plasma with the use of chromatographically clean CO in the amount 0.8 cm^3^/50 mL plasma. The study of the animals was planned on the basis of the methods included in the publication by Koziorowski [22] with modifications.

All surgical procedures of cannula insertion to the jugular vein and angular vein through the dorsal nasal vein were performed under general anaesthesia.

A schematic drawing of the experiment is in the Appendix A.

### 2.2. RNA Analysis

The RNA isolations were performed on the retinal tissues of post-mortem animals with the use of the TRI reagent (Merck, Kenilworth, NJ, USA). The isolation kit was based on an improved method developed by Chomczynski [23]. The obtained RNA was used for reverse transcription into cDNA, using the high-capacity cDNA Reverse Transcription Kit (Thermo Fisher Scientific, Waltham, MA, USA). The real-time PCR method was used to study gene expression. Real-time PCR was achieved with the use of StepOne Plus^TM^ Real-Time PCR Systems (Applied Biosystems, Waltham, MA, USA). The reaction mixture contained Taq Man Universal Master Mix II (Thermo Fisher Scientific, Waltham, MA, USA), specific assay Id: ss03376410—*HO2*, ss03378516—*HMOX1*, ss03375629—*GAPDH*, ss03377029—*PCNA* (Thermo Fisher Scientific, Waltham, MA, USA). In the reaction, 50 ng of cDNA was used. GAPDH (glyceraldehyde-3-phosphate dehydrogenase) was used as a housekeeping gene. The gene expressions in different samples were compared using the ΔΔCt method.

### 2.3. Protein Analysis (Western Blot Method)

Protein was isolated with the use of a Fast-trap device equipped with tubes filled with zirconium balls (1.0–1.2 mm) and 2% SDS (Merck, Kenilworth, NJ, USA). Supernatants containing protein were obtained through centrifugation at 14,000 rpm for 15 min at 4 °C. The concentration of protein was determined by using the BCA method (Thermo Fisher Scientific, Waltham, MA, USA). The experiments were conducted in 10% SDS-PAGE. The proteins were then transferred to the nitrocellulose membrane that was blocked with 5% milk powder (degreased) in TBS-T (20 mM Tris (Merck, Kenilworth, NJ, USA), 137 mM NaCl (Chempur, Piekary Śląskie, Poland), 0.1% (*v/v*) Tween 20 (Merck, Kenilworth, NJ, USA), pH 6.6) for 1 h at room temperature. The specific antibodies were rabbit polyclonal anti-Actin (1:5000 diluted), mouse monoclonal anti-HO-1 (1:500 diluted), goat polyclonal anti-HO-2 (1:500 diluted) (Thermo Fisher Scientific, Waltham, MA, USA) and mouse monoclonal anty-PCNA (1:1000 diluted) (Santa Cruz Biotechnology, Dallas, TX, USA). The antibodies were incubated with the membrane overnight at 4 °C. The secondary antibodies were goat (1:5000 dilution) (Santa Cruz Biotechnology, Dallas, TX, USA), anti-rabbit IgG- (1:40,000 dilution) (Merck, Kenilworth, NJ, USA) and anti-mouse, which were incubated for 1 h at room temperature [24]. The protein bands were visualised with Fusion FX7 Vilber film imaging.

### 2.4. Statistical Analysis

The results are presented as mean ± SEM from at least three independent experiments. Statistical assessment was performed through analysis with one-way ANOVA (Bonferroni: compare all pairs of columns) analysis for three group comparisons using Graph Pad Prism 9 (GraphPad Software, San Diego, CA, USA). Differences were accepted as statistically significant at *p* < 0.05.

## 3. Results

The observed difference in the expression of *HO-1*, *HO-2* and *PCNA* gens among the three groups of studied animals (Figure 1).

The most significant changes were observed for the expression of the *HO-2* gene. The statistical significance for the tested animal groups were *p* < 0.05 (for groups with CO and without CO) and *p* < 0.001 (without CO and control). Expression of the *HO-2* gene was highest for animals who received CO by infusion (1.108 ± 0.095) and noticeably lower for animals who received plasma without CO (0.737 ± 0.037) compared with the control (Figure 1b). The expression of the *HO-1* gene was highest for examined animals who received CO by infusion (1.282 ± 0.154) and slightly lower for animals who received plasma without CO (0.978 ± 0.113) compared to the control (1.0 ± 0.116) (Figure 1a). *PCNA* gene expression for animals with plasma infusions (1.199 ± 0.164) and plasma with CO infusions (1.18 ± 0.128) increased compared to the control (Figure 1c).

All the protein levels are presented as means ± SEM (normalised to the β-actin). The obtained results are presented in Figure 2.

The most statistically significant results were observed for HO-1 and HO-2 proteins (Figure 2). The largest decrease in protein level was observed for HO-1 in animals with plasma infusion without CO. The HO-1 protein concentration was at (0.036 ± 0.009) compared with the control (1.0 ± 0.159), and the statistical significance between these groups of animals was *p* < 0.001. The animals with CO-enriched plasma infusion had a higher HO-1 protein concentration (0.802 ± 0.196) than the animals infused without CO (0.036 ± 0.009), but this was lower than animals from the control group. The statistical significance between CO-infused animals group and the without CO group was *p* < 0.01 (Figure 2a). We observed a similar situation for the HO-2 protein concentration in the retina. The lowest concentration was observed in animals with plasma without CO infusion (0.071 ± 0.017 and *p* < 0.001 compared with control), and a higher concentration in animals with plasma enriched with CO infusion (0.382 ± 0.088 and *p* < 0.001 compared with control), but these results were below the control (1.00 ± 0.146). The statistical significance between animal groups with CO infusion and those without CO infusion was *p* < 0.05 (Figure 2b).

Changes were also observed in the PCNA protein concentration. Compared with the control, the lowest concentrations of protein were observed in animals with plasma infusion without CO (0.533 ± 0.085, *p <* 0.058, compared with control), and in this case we can see the trend was very close to statistical significance. However, higher concentrations were observed in animals infused with CO (0.996 ± 0.239) in comparison with the control (1.00 ± 0.096) (Figure 2c).

## 4. Discussion

Carbon monoxide is a toxic gas, but we have highlighted that it is also an intracellular neuromodulator or neurotransmitter, the same as nitric oxide (NO) [25]. In earlier studies, we examined HO enzyme changes in the retina in two points of the year with the highest and lowest sunlight intensity [26]. In the presented research, we checked how the examined factors of HO-1, HO-2 and PCNA changed in the retina after exogenous CO concentration increased in the blood outflow from the eye due to a lack of light.

The CO particles carry information about changes in light intensity by the humoral pathway from the eye to the hypothalamus and pineal glands, using the venous ophthalmic sinus and the further cavernous sinus through the transport particles to the rete mirabile carotis [27]. These structural connections of blood vessels in the brain confirm our hypothesis. Our previous study showed that endogenous CO is elevated physiologically on long days in blood, which flows out from the eye, and is three time higher than at night. We also observed elevated CO in blood outflow from the eye on short days compared to blood outflow from the nose as a control [22]. These results suggest that blood flows through the thin-walled vessels in the retina with a substantial stock of substrate, because the enzymatic transformation of heme oxygenase depends on light to biliverdin and CO [28], causing an increase in CO in the blood. Further studies revealed that pig retina are highly sensitive to changes in daytime. We observed a significant increase in mRNA *HO-1* and *HO-2* on short winter days, in comparison with that on long summer days. Further analysis proved that light-dependent expression was demonstrated by the increased translated protein of examined genes *HO-1* and *HO-2* in long daytime, unlike short daytime [26]. 

This paper’s results confirm the previous data presented above. An abundance of *HO-1* and *HO-2* mRNA in the retinal tissues of the examined pigs showed there were only significant differences in HO-2 between the groups. However, these changes are related to the decreased mRNA expression of *HO-2* in the group of animals housed in constant darkness without the addition of CO. An addition of CO caused elevated RNA expression of all examined genes, but only *HO-2* showed statistically significant changes compared with the control group housed in natural conditions. The presented results are consistent with those of Kim et al. which showed that CO upregulates *HO* expression [10]. In this study, we reveal that light-dependent changes appear on a translational level. Each examined factor decreased its own protein expression in the group housed in constant darkness without the addition of CO, but the addition of CO to infused plasma caused a statistically significant elevation in HO-1 and HO-2 protein expression, and increase in PCNA. Even though the HO-1 and HO-2 levels did not increase to the protein level in the normal group, we still observed a several-fold higher concentration of protein HO-1 and HO-2 in the group with the addition of CO compared to the group without CO. This data confirmed our hypothesis about the role of CO in light transmission, and we realize that CO only modulates and supports light because it is necessary to produce this particle. However, as described above, the addition of exogenous CO modulates the physiological regulation on the translational level of heme oxygenases. Moreover, our observations indicate that the translation of HO-1 and HO-2 is upregulated by CO. Endogenously synthetized CO in retinal tissue is more available because CO is produced locally. Assuming the presented data are consistent, they help us to determine what process can be triggered by CO between mRNA transcription and protein translation.

Exogenous CO as an additive was also used in wild boar–domestic pig crossbreeds housed in natural conditions and examined after the animals were infused on long and short days. We observed that CO reached the hypothalamus and pineal gland in the brain and modulated many physiological pathways including the molecular clock and gonadotropins synthesis in the hypothalamus and melatonin synthesis in the pineal gland. We know that CO dysregulates the intrinsic biological clock in the preoptic area (SCN) and dorsal part (PVN) of the retino-hypothalamo-pineal pathway (RHP) in the hypothalamus [29]. The observed changes in the gene expression of *BMAL1*, *NPAS2* and *CLOCK* as transcription factors of *Per 1-2, Cry 1-2* and *Rev-erb α* and *β* occur in both structures after the addition of CO [14]. We also examined *GnRH* expression in the hypothalamus and observed significant changes in gene expression and hormone levels (FSH, LH) after the addition of CO compared with control group [15]. Finally, our team examined the enzymes of the melatonin pathway in the pineal gland, where we also observed changes in the gene expression of *AANAT* and *HIOMT*, especially on a protein level after exogenous CO was added [18].

Furthermore, the activation of HO expression at the mRNA and protein level in the retina may have broader implications for the functioning of this part of the nervous system. Previous studies have shown that the CO-HO circuit and exogenous CO exert positive effects on the nervous system’s functioning. CO was shown to act on the transient activation of HO-1, stimulating regenerative processes in the acute ischemic injury of the CNS. This regenerative function is supported by another CO-HO circuit function, either neuroinflammation suppression or neurogenesis promotion [20,30].

The studies on *PCNA* gene expression showed no differences among groups of animals. Differences were observed in the amount of PCNA protein in the individual groups. The relationship between changes in the amount of PCNA protein was consistent with the amount of HO proteins isolated from the retina in each group. This shared relationship is related to the presence of CO. 

Summarizing our current and previous research as common results, all experiments have shown consistent results, despite being conducted at different times, in different places, and on different animals. The previous and current results have shown us:-Carbon monoxide in small quantities is not toxic to organisms but has a stimulating function;-Carbon monoxide modulates the expression of studied factors (*HO-1, HO-2* and *PCNA*), e.g., humoral light signal;-CO synthesis in natural conditions upregulates itself and depends on the amount of particles which are consistent with light intensity during the different seasons;-The amount of HO-1 and HO-2 proteins in the retina is dependent on the CO concentration in the blood flowing to the eye;-Many of the changes we have observed are on a translational level, and we conclude that CO particles regulate cofactor-triggered translation.

## 5. Conclusions

Our data demonstrate that exogenous CO and CO-HO’s circuit are important for the proper functioning of retinal tissue in both natural conditions and constant darkness. We also showed that the humoral pathway for light transmission is an efficient modulator of many neural systems.

## Figures and Tables

**Figure 1 biomedicines-10-00358-f001:**
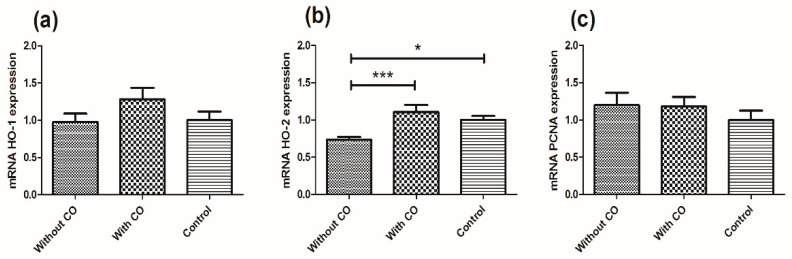
Expression of HO genes (HO-1 (**a**) and HO-2 (**b**) and PCNA genes (**c**) in the retina of the domestic pig (*Sus scrofa f. domestica*) in three research groups: plasma infusion without CO, plasma infusion with CO (0.8 cm^3^ CO for 50 cm^3^ plasma) and control group not infused with plasma; statistically significant at: *p* < 0.05—*; *p* < 0.001—***.

**Figure 2 biomedicines-10-00358-f002:**
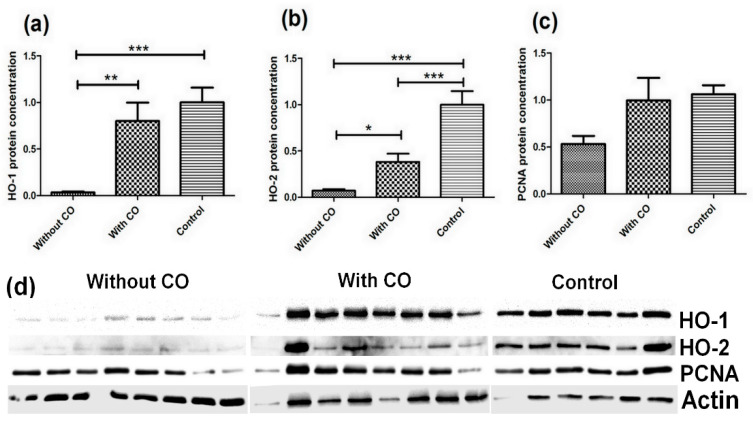
Protein concentration of the HO-1 enzyme (**a**), the HO-2 enzyme (**b**), and the PCNA protein (**c**) in the retina of the domestic pigs (*Sus scrofa f. domestica)*. The material was divided into three groups: plasma infusion without CO, plasma infusion with CO (0,8 cm^3^ CO for 50 cm^3^ plasma) and control without plasma infusion. The protein level was determined with the Western blot method, and the densitometry analysis was performed after normalisation with β-actin (**d**) immunoblotting with a specific antibody; statistically significant at: *p* < 0.05—*; *p* < 0.01—**; *p* < 0.001—***.

## Data Availability

All data are available from corresponding author (S.N.). Animal experimentation data (detailed animal model) are available from the corresponding author (P.G.).

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
