# Peer review of "Carbon Monoxide (CO) as a Retinal Regulator of Heme Oxygenases -1, and -2 (HO’s) Expression"

_biomedicines, 2022, doi:10.3390/biomedicines10020358_

Round 1

Reviewer 1 Report

That CO is a direct regulator of HO's is known, previously demonstrated in vitro in hepatocytes and endothelial cells. However, in the current manuscript, for the first time, the authors show that CO (as a light signal) directly modulates the expression of HO-1 and HO-2 in vivo in the retinae of pigs. The message of the study is clear and as noted by the authors, the study has broad applications in other tissues and systems of the body especially regarding potential tissue regenerative strategies. Additionally, the study was executed with very fine experimental strategy and techniques. Notwithstanding, the following are my concerns which if possibly addressed may improve upon the quality of the current manuscript.

Major concerns:

  1. Immunohistochemistry (IHC) for HO-1 and HO-2 in the retinae of the 3 study groups is suggested. The retina is multicellular, and IHC will help elucidate relative cell-specific localization of the HO's.
  2.  If possible, colocalization of the HO's with markers for retinal vasculature, neurons, and/or glia in retinal cross-sections should be done via IHC.
  3.  The current study lacks possible mechanism (s) underlying CO-induced modulation of the HO's. At least, performing Western blots for already purported CO-HO-mediated mechanisms in the literature would be helpful in revealing an associated mechanism.

Minor Concerns:

  1. The entire manuscript should be edited for clarity, text editing, and grammar.
  2. The introduction and discussion are loaded with important information. They however appear to be long and being succinct and concise may help to cut down on their respective length.
  3. Although the authors have described some of their methods in detail in other published papers like the Gilun et al. (2021) in IJMS, it would be helpful in the current manuscript to sufficiently describe in detail the methods (together with a possible summary schematic).
  4. The authors should be clear in the results and discussion sections of their paper in presenting findings that had an upward trend yet not statistically significant. I suggest they tone down on discussing results that had an upregulation trend, yet not statistically significant.
  5. I'm curious knowing the authors' thoughts on their experimental findings if they had included pigs under normal environmental (light/dark) conditions infused with CO as one of the study groups.
  6. Do the authors possibly know the normal diurnal rhythmic levels of CO in pigs under normal environmental conditions?

Author Response

Thank you for your valuable/constructive comments in your review.

Major concerns:

  1. Immunohistochemistry (IHC) for HO-1 and HO-2 in the retinae of the 3 study groups is suggested. The retina is multicellular, and IHC will help elucidate relative cell-specific localization of the HO's.
  2. If possible, colocalization of the HO's with markers for retinal vasculature, neurons, and/or glia in retinal cross-sections should be done via IHC.

Answer:

The study was treated as preliminary research, the results of which will enable us to apply for a grant for research that will broadly elucidate the mechanisms by which carbon monoxide is involved in the functioning of the biological clock in the hypothalamus. Therefore, we decided to present these preliminary results in the form of a Communication in a Special Issue of this journal. Tissue samples have not been adequately fixed, so it is not possible to perform IHC analysis (and/or colocalization of analyzed factors) at this time. This is certainly a very valuable observation and such analyses will be performed in planned future studies.

Rewiever:

   The current study lacks possible mechanism (s) underlying CO-induced modulation of the HO's. At least, performing Western blots for already purported CO-HO-mediated mechanisms in the literature would be helpful in revealing an associated mechanism.

Answer:

As mentioned above, we consider the presented results as preliminary studies, while CO-OH-mediated mechanisms studies are planned for the future. Currently, we do not have enough collected tissues to perform additional western blot analyses. We will use these valuable comments/suggestions in planning future studies.

Minor concerns:

  1. Text was proofread by native spell checker.
  2. language corrections have been made to simplify as much as possible without changing the content.
  3. We attach to the supplementary data an outline drawing with a diagram of the experience. We intentionally do this because, as mentioned above, this is a communication type paper and we do not want to exceed the limit of 4000 words.
  1. in the results section an increasing trend for PCNA protein level is included because the p-value<0.058 indicates that with slightly more animals this value would be statistically significant.
  2. such experiments were done where under natural conditions animals were given CO infusion for 48 h, however, we did not examine the retina itself but focused on changes in the hypothalamus and pineal gland. Citations: 14, 15, 18, 22.
  3. We are familiar with the parameters of diurnal variation of endogenous CO in the blood draining from the eye during the short and long day seasons, and these data are published in Koziorowski et al. 2012, entry: 22

Best regards,

Reviewer 2 Report

The manuscript entitled “Carbon monoxide (CO) as a retinal regulator of heme oxygenases-1, and -2 (HO’s) expression” describes the study of the relationship pf CO concentration with the three HO enzymes expression in an in vivo study on domestic pig eye. The authors divided 22 animals in three experimental groups, infused plasma CO depleted and kept in the dark; infused plasma with increased CO and kept in the dark; and control group without plasma infusion and kept under natural light conditions. After the experiment was done, animals were sacrificed, and the pig’s retinas were studied by means of RNA isolation and posterior RT-PCR; and Western Blot protein analysis. The results show a statistical decrease in HO gene expression and HO protein concentration found in the experimental group deprived of CO.

The manuscript is well written and easy to understand. The experiments carried out are adequate for their purpose of the investigation. The references used in the manuscript are recent and are adequate.  Regarding the novelty of the manuscript as far as I am concerned there are some works on the topic performed by the same research group, but as the authors state this is the first time that the HO gene expression and HO protein concentration studied in these experimental groups.

In my opinion the results shown in the present manuscript are interesting for a broader community.  Despite its great potential, the paper comes with some minor issues which are addressed below:

  • Revise English, there are sentences which lack the verb, others lack adverbs.
  • Page 2 line 23: affects for iron metabolism à affects iron metabolism
  • Page 2 line 24: ant à and
  • Page 2 line 45: It non-physiological changes à Non-physiological changes
  • Some citations on the text have mixed criteria: The study of the animals was planned on the basis of the methods included in the publication by Koziorowski et al. 2012 [22]. You should delete et al. and the year. Same happens with Gilun et al 2021 [21] and Chomczynski 1993 [23].
  • Page 3 line 38: define RT-PCR, it is defined in same paragraph line 40, also the word order is inverted PCR Real Time àReal Time PCR.
  • Page 3 line 50: Separate number from units: (1.0-1.2mm) à (1.0 – 1.2 mm).
  • Page 4 line 1: Separate the degree from the number: 4°Cà4 °C.
  • Page 4 line 26: I would change patients to animals.
  • Change statistical p values to italics.
  • Change all decimal commas in pages 4 and 5 to decimal points.
  • Use italics on gene names.

Best regards

Author Response

Thank you very much for your insightful review of the publication and your valuable comments. All suggestions have been applied in the text. After changed, text was proofread by native spell checker.

Best regards,

Round 2

Reviewer 1 Report

The authors have adequately justified why suggested experiments cannot be done at this time.